# A Study on Distance Measurement Module for Driving Vehicle Velocity Estimation in Multi-Lanes Using Drones

Kwan-Hyeong Lee

Division of IT Convergence, Daejin University, Pocheon 1007, Korea; khlee@daejin.ac.kr

**Abstract:** A method of estimating driving vehicle information usually uses a speed gun and a fixed speed camera. Estimating vehicle information using the speed gun has a high risk of traffic accidents by the operator and the fixed speed camera is not efficient in terms of installation cost and maintenance. The existing driving vehicle information estimation method can only measure each lane's driving vehicle information, so it is impossible to measure multi-lanes simultaneously with a single measuring device. This study develops a distance measurement module that can acquire driving vehicle information in multi-lanes simultaneously with a single system using a drone. The distance measurement module is composed of two LiDAR sensors to detect the driving vehicle in one lane. The drone is located above the edge of the road and each LiDAR sensor emits the front/rear point of the road measuring point to detect the driving vehicle. The driving vehicle velocity is estimated by detecting the driving vehicle's detection distance and transit time through radiation, with the drone LiDAR sensor placed at two measurement points on the road. The drone LiDAR sensor radiates two measuring points on the road and estimates the velocity based on driving vehicle's detection distance and driving time. As an experiment, the velocity accuracy of the drone driving vehicle is compared with the speed gun measurement. The vehicle velocity RMSE for the first and second lanes using drones is 0.75 km/h and 1.3 km/h, respectively. The drone and the speed gun's average error probabilities are 1.2% and 2.05% in the first and second lanes, respectively. The developed drone is more efficient than existing driving vehicle measurement equipment because it can acquire information on the driving vehicle in a dark environment and a person's safety.

**Keywords:** distance measurement module; vehicle velocity estimation; drones; LiDAR



## 1. Introduction

Drones can be applied to detect a wide variety of objects. The drone's object detection methods involve global positioning system (GPS) jamming, radar, radio wave signal detection (radio frequency sensing), image cameras and light detection and ranging (LiDAR). The GPS jamming method [1] can disable the GPS safety mode by emitting a fake GPS signal to a nearby object, converting the object path and making the current position misunderstood. However, this method is not legally permitted outside of the exhibition situation and specific time points. The radar method [2,3] detects the object by extracting information from the received signal by radiating radio waves, but the resolution is not better than other methods. The radio wave signal detection [4,5] finds both the drone and the manipulator by receiving the radio wave used by the drone without emitting radio waves like the radar and acquiring the frequency used for communication with the drone controller. However, this method's problem is to find out the presence or absence of radio waves and the specific direction of the drone cannot be known. In addition, it is challenging to find drones if they fly autonomously without communication. In the method of detecting an object using an image camera [6,7], the object is identified, but the ability to detect the object is degraded in rain, snow and dark environments. The LiDAR method [8–11] detects the object by analyzing the object's received signal by emitting the

transmission signal with a small beam width. This method has a higher object detection accuracy than other methods.

Giuseppina et al. [12] studied the lane support system to prevent accidents caused by road departure and used a decision tree method to analyze the cause of the defects and the importance of the variable involved in the process. Marek et al. [13] studied the method of identifying the location of garbage from images collected using the drone camera and displaying it on a global map using the on-board sensor set. Hyeon et al. [14] studied a hardware architecture applicable to a system-on-chip and a vision-based tracking algorithm to track objects in the drone hovering state. Muhammad et al. [15] studied 3D maps using the data acquired from the drone. Ruiqian et al. [16] studied the method for optimizing the resolution of images detected by the drone. This method improved the image resolution extracted from the existing object detection network by applying an extended convolutional network to the global density model of the drone's object detection. Victor et al. [17] studied a method to improve the above-ground biomass estimation performance by attaching a LiDAR sensor and a multispectral camera to an unmanned aerial vehicle platform. To compare the drone LiDAR data with the above-ground biomass estimate in the same area, the point cloud was created by applying a red-green-blue (RGB) image and a structure for the motion process and the normalized difference vegetation index of the multispectral image was calculated. Srinvasa et al. [18] studied autonomous vehicles to detect and avoid obstacles. In this method, a mapping technology that detects obstacles and creates an image of the surrounding environment using the Raspberry Pi and the LiDAR module without the computer vision technology was examined. Razvan et al. [19] studied a method in which a vehicle detects objects, obstacles, pedestrians or traffic signs in foggy weather conditions by using Laser and LiDAR methods to estimate driving visibility. This method determines the vehicle operation method according to the vehicle and improves the autonomous vehicle's stability. Charles [20] presented two methods to extract vehicle velocity. The first method was to extract the vehicle velocity by representing the 3D coordinates of the vehicle point with two images acquired with a static camera and minimizing residual errors and dimensional deviations. The first method's velocity extraction performance was compared with the second LiDAR method for the vehicle velocity measurement. Takashi et al. [21] studied a pedestrian recognition algorithm that detects pedestrians using high-resolution vehicle LiDAR sensors and adapts them to traffic congestion as a method to reduce pedestrian accidents. Fernando et al. [22] detected a vehicle using a laser radar mounted on the vehicle bumper in a road environment. This method estimates vehicle movement and shape by fusion of computer vision and laser radar data. Donho et al. [23] studied object detection and tracking using a camera mounted on the drone. The motion compensation of the drone used frame subtraction, morphological operation and false blob removal. However, an error occurs in the position and the object's velocity if the sampling time is not accurate and the velocity of the object is slow. Jianqing et al. [24] studied ground filtering and point clustering techniques in windy weather by attaching light detection and ranging to a fixed device and compared the performance with existing data processing algorithms. Shuang et al. [25] studied the object matching framework based on affine-function transformation for vehicle detection from the crewless aerial vehicle's camera image. This method effectively handles vehicles in various conditions such as scale change, direction change, shadow and partial occlusion.

The driving vehicle detection is usually measured using the speed video camera of a gentry structure installed on the road or the speed gun at the edge of the road since the driving vehicle information detector of the gentry structure is not efficient for mobility because it is fixed and the method of detecting information on the driving vehicle using the speed gun on the road's outskirts has a high risk of traffic accidents for the measuring person.

In Figure 1, the vehicle velocity estimation is performed by a person using the speed measuring device on the road's outskirts. When a measurer detects the vehicle information on the road's outskirts, it is impossible to secure stability against traffic accidents. In addition, when estimating the driving vehicle information using the vehicle speed measuring

device on the outskirts of the road, it is difficult to estimate accurate information because the position and inclination of the measuring device are changed due to wind resistance.

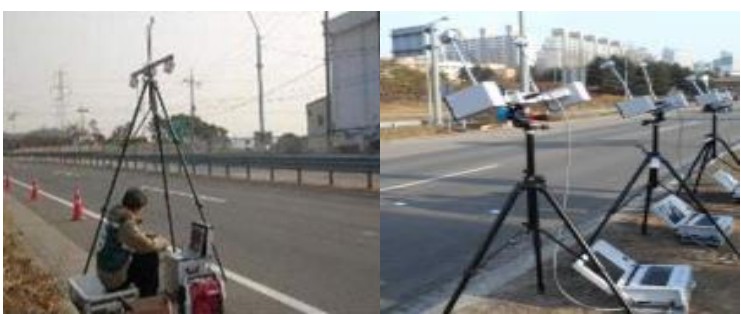

**Figure 1.** Vehicle velocity measurement with existing equipment.

In this study, the drone distance measurement module is developed to obtain driving vehicle information with safe and efficient mobility. The drone distance measurement module measures the distance between the driving vehicle and the drone and the vehicle passing time between two measurement points on the road.

Figure 2 shows the road driving vehicle information acquisition system using the drone developed in this paper. The vehicle information acquisition system consists of a drone vehicle detection part (airborne equipment) and a ground control part (ground equipment). The drone vehicle detection part is composed of the LiDAR sensor, vehicle detection module (VDM) and detection information acquisition module (DIAM), whereas the ground control part is composed of the detection module control analysis (DMAC) to monitor and control the state of the drone and the detection information acquisition analysis (DIAA) to calculate the vehicle velocity. The drone vehicle detection part indicates two specific points on the road lane using the LiDAR sensor installed on the drone. The measuring point is set as a front point and a rear point for two points on the road, while the drone measures the distance between the drone and the measurement point when the vehicle passes through two points. The drone vehicle detection module is composed of two LiDAR sensors in one lane to obtain vehicle information. It is composed of four LiDAR sensors to obtain driving vehicle information in two lanes. The drone's entire vehicle detection module developed in this study is composed of six LiDAR sensors to acquire vehicle information driving in multi-lanes.

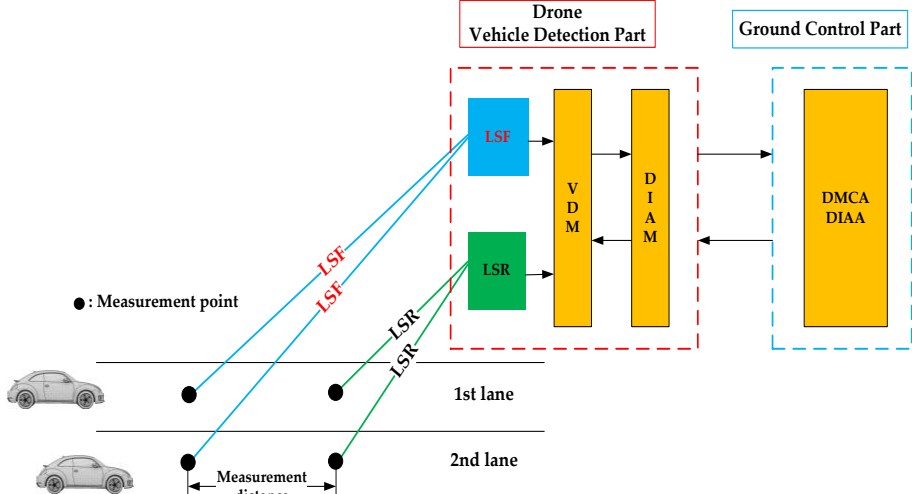

**Figure 2.** Vehicle information acquisition system: LSF—LiDAR sensor front; LSR—LiDAR sensor rear; VDM—Vehicle detection module; DIAM—Detection information acquisition module; DMCA—Detection module control analysis; DIAA—Detection information acquisition analysis.

The drone vehicle distance measurement method sets the distance between the drone and the measurement point by indicating the road's front and rear points by two LiDAR sensors mounted on the drone. The distance between the LiDAR sensor and the measuring point changes when the vehicle passes the measuring point. The distance between the LiDAR sensor and the measuring point changes when the vehicle passes through the two measuring points.

## 2. Distance Measurement Module Development

### 2.1. Vehicle Detection Module Analysis

The vehicle detection module acquires information using LiDAR sensors at two measuring points in the laser for the vehicle detection and the velocity calculation and transmits the information to the ground control part. The acquired information is essential information for detecting the vehicle and determining the velocity. The vehicle detection module is necessary to detect the driving vehicle, change the angle for each sensor, check the sensor arrangement and transmit data. The sensor angle change and the sensor placement determine vehicle velocity for two points on the road.

Figure 3 is a connection diagram of the ground control part and the drone vehicle detection part. The drone vehicle detection part is composed of VDM, DIAM, camera and storage medium. The VDM creates raw sensor information for the vehicle detection and velocity calculation in the measurement range. The VDM collects the detection sensor information of two measuring points to calculate the vehicle velocity and transmits it to DIAM. The collected sensor information is important data for driving the vehicle detection and the velocity calculation. The VDM functions for detecting the driving vehicle are as follows:

- detection of the driving vehicle distance between two measuring points;
- change of LiDAR sensor angle to measuring point distance;
- check the LiDAR sensor placement;
- the function of transmitting and processing the collected raw data.

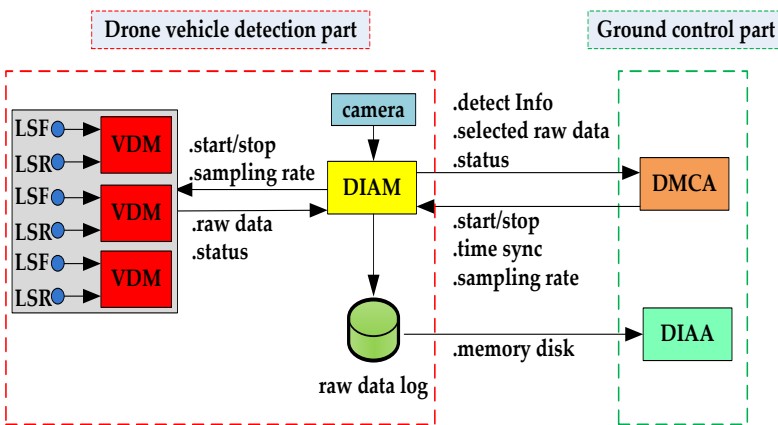

**Figure 3.** Vehicle information detection system.

The VDM requires two LiDAR sensors to generate raw data for detecting the vehicle in each lane and a laser indicator to check the location of the LiDAR sensor projecting. The VDM communicates with the DIAM using three universal asynchronous receiver transmitters (UARTs).

The VDM and DIAM of the vehicle detection part transmit data and control signals using serial communication and the camera and DIAM transmit information using high-definition multimedia interface (HDMI) communication. The 915 MHz radio frequency (RF) communication is used transmit control signals and data between the drone vehicle detection part and the ground control part. The LiDAR sensor front (LSF) and LiDAR sensor rear (LSR) are front/rear LiDAR sensors at the road's vehicle measurement point. The VDM is a device for detecting the driving vehicle and requires one VDM in one lane.

Three VDMs are required to detect all vehicles driving in three lanes and one VDM is equipped with two LiDAR sensors. Vehicle information and the VDM status collected from the LiDAR sensor are transmitted to DIAM. The DIAM transmits start and stop signals and sampling rate to the VDM for the vehicle detection. The DIAM collects each lane's sensor distance information as measured by the camera and the VDM and stores the raw data in the vehicle detection and the storage medium (micro san disk memory). The DIAM transmits data to the ground control part the DIAA to calculate traffic volume and vehicle velocity in real-time. The ground part DMCA calculates the vehicle velocity using the DIAM data and the DIAA extracts the vehicle detection and the velocity using the raw information of the DIAM storage medium to extract more accurate traffic information.

### 2.2. Vehicle Detection Module Data Flow

The distance measurement module collects information on the driving vehicle by connecting to two LiDAR sensors that detect two measuring points in the lane. The collected raw information is encapsulated and transmitted to the DIAM. In addition, it should be possible to control and set the operation of the LiDAR sensor for the DIAM command signal. Therefore, the VDM control specifications for the distance measurement module are as follows:

- use of three or more communication ports;
- 32 kbps or higher communication speed specification;
- circular queue that can store 100 data per communication port;
- communication port 2000 bytes per second interrupt processing capacity;
- sensor information parsing function;
- framing function for the DIAM transmission;
- sensor control function and status check function.

Figure 4 shows the vehicle detection flow chart of the VDM. The light emitting diode (LED)/watchdog/LiDAR sensor periodically checks the hardware status of the VDM and communicates data processing to the UART. The data processing receives raw data and the vehicle detection status from periodic and beaglebone black communication (BB comm). Then, it transmits the data to the gate keeper in the form of a frame. In addition, the measured data from the LiDAR sensor is converted into distance information and stored as a queue. The BB comm controls each lane measurement LiDAR sensor to analyze information received from the DIGM using UART. The gate keeper sequentially transmits information to the DIAM from the distance information queue.

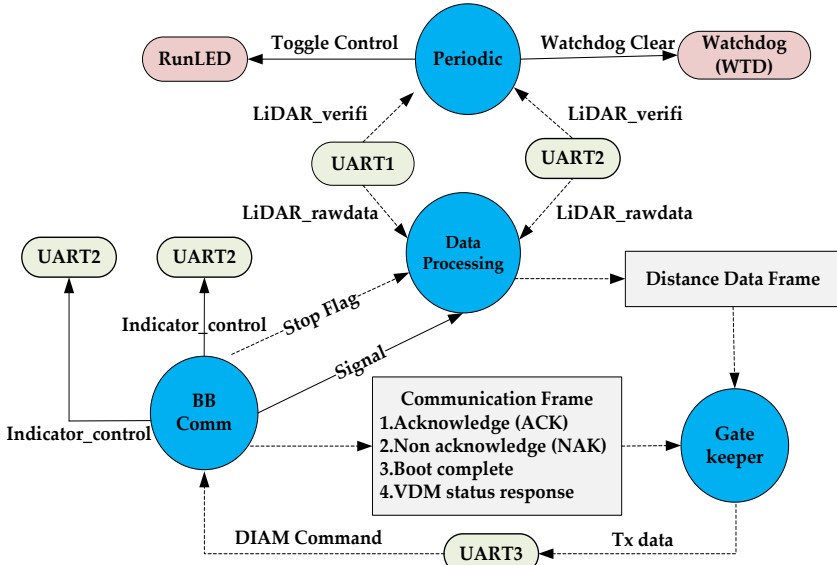

**Figure 4.** Vehicle detection module (VDM) data flow diagram.

*2.3. Vehicle Detection Module Design and Development*

The VDM connects two LiDAR sensors for the front/rear measurement points of the lane to generate raw vehicle information for each lane and the laser indicator is used to check the measuring point of the LiDAR sensor. The VDM uses three UARTs to communicate with the DIAM and requires general purpose input output (GPIO) for LED and laser indicator control. The drone developed in this paper is located above the outside of the road and detects the driving vehicles from 1st to 3rd lanes by hovering. The drone is designed to have a detection error of about 3% or less when the vehicle's maximum driving velocity is 170 km. The factor having the most significant influence on the accuracy of vehicle detection information is sensor performance (collection cycle/distance measurement) and the sensor accurately indicates the point of the road measurement distance. It is crucial for each sensor to precisely indicate the lane measurement point to collect accurate vehicle information. In this paper, a high-power laser indicator is used to confirm that the LiDAR sensor accurately indicates the lane's measurement point and the vehicle information of each lane is detected by controlling the indicator on/off. Figures 5 and 6 reference the VDM hardware artwork and the VDM printed circuit board (PCB) developed in this study, respectively. Figure 5 uses a program (Or-Cad) to design a circuit and PCB is manufactured as an outsourcing service. The VDM is connected to two LiDAR sensors that detect two-lane points and transmits the encapsulated raw information to the DIAM. The VDM should control and set the sensor operation according to the command received from the DIAM; that is, the VDM transfers the collected information to the DIAM by being connected to two sensors that detect lane measurement points, with the VDM operating the sensor according to the DIGM command.

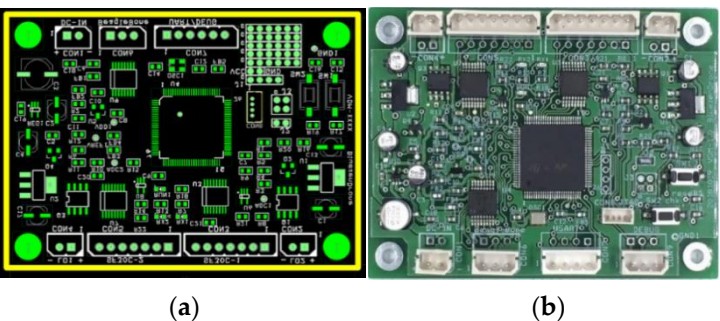

| (**a**) | (**b**) |

**Figure 5.** This is a VDM of the vehicle detection module part: (**a**) VDM hardware (H/W) artwork; (**b**) Developed VDM.

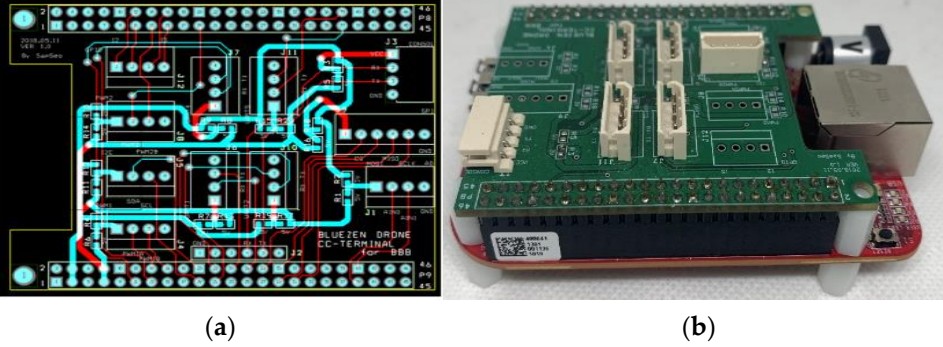

| (**a**) | (**b**) |

**Figure 6.** This is a detection information acquisition module (DIAM) of the vehicle detection module part: (**a**) DIAM H/W artwork; (**b**) Developed DIAM.

The VDM control specification is as follows:

- three or more communication ports (UART);
- 32 kbps or higher communication speed;

- circular queue capable of storing 1000 data per communication port;
- interrupt processing capacity of more than 2000 bytes per second of communication port;
- sensor information parsing function and farming function for the DIGM transmission;
- sensor control function and status check function.

### 2.4. DIAM Design and Development

The DIAM is saved in real-time to collect the sensor distance information of each lane from three VDMs. The stored data are transmitted to the DMCA of the ground control part in real-time to calculate the traffic and the velocity information. The real-time raw information is stored in the micro san disk (SD) memory unit and the DIAA of the ground control part measures the accurate vehicle information when the traffic information collection is completed. The detection algorithm determines the vehicle front/rear using the collected raw information. The generated information is transferred to the DMCA queue.

The vehicle is detected using the detection algorithm and the DMCA transmission is determined by the command mode received from the DMCA detection mode. The detection algorithm is composed of normal mode and debug mode. The normal mode transmits the vehicle entry/exit detection information to calculate the vehicle velocity in the DMCA. The information of each sensor is checked in debug mode. In this mode, the DMCA sends 1 or 2 sensor selection commands and the selected sensor information is grouped by 10 to be transmitted to the DMCA.

➢ Normal mode activity:
- sensor number: check the sensor status detected entry/exit.
- status: front/rear status;
- tick: calculation of the time required for tick data rate in the front/rear detection (operates at 800 us/tick);
- base value: represents data to obtain the distance between two sensor measurement points;
- data produce: all distance information received from the VDM is saved in a file.

➢ Debug mode activity:
- sensor number: identify and check the selected sensor information;
- current value: check the current sensor distance data;
- base value: check the currently applied base value;
- status: check the current status of sensors such as vehicle entry/exit, standby, status, etc.;
- wait count: a specific time is counted when the condition of transitioning from the current stat to another state is satisfied (800 us/count).

Figure 6 shows the DIAM artwork and the developed DIAM (Figure 6); DIAM processes more than 1000 distance information per second from the three VDMs simultaneously and transmits it to the DMCA.

Figure 6 the DIAM performs the following functions:

➢ VDM and communication function (RS232 communication):
- acquisition of front/rear real-time sensor distance information;
- acquisition of sensor operation status information;
- control commands such as start/stop of collection;
- simultaneous processing of multiple VDM information.

➢ DMCA and communication function (Ethernet 915 MHz wireless):
- compression function of real-time detection information (transmission of entry/exit event information);
- integrity of event information delivery (transmission confirmation and error check);
- real time information transmission function for validating detection processing.

➢ Vehicle detection function:
- distance value slope and change amount search function;

- front/rear decision;
- determine the validity of status transition;
- separate management of parameters for each sensor (expected to be different for each detection location and inspection pattern).
➢ Real-time raw information storage function:
  - include in the parameter file header for vehicle detection.

### 2.5. Velocity Estimation

This study develops the VDM and the DIAM to mount the drone as mission equipment. The driving vehicle information of each lane is obtained by hovering the drone at the outside the lane. Figure 7 shows the vehicle detection method using the drone. The drone LiDAR sensor radiates two measuring points to obtain information on the vehicle driving in the lane. $H_f$. is the distance between the entrance measurement point and the LiDAR front sensor, $H_r$ is the distance between the LiDAR rear sensor and the exit measurement point, $\theta_s$ is the angle between the front/rear sensors and $L$ is the distance between the two measurement points. The two LiDAR sensors are assigned to one lane and each sensor detects the vehicle by radiating each front/rear point. The distance ($L$) between the two measurement points is changed according to the flying height of the drone and the distance is calculated by Equation (1) [26].

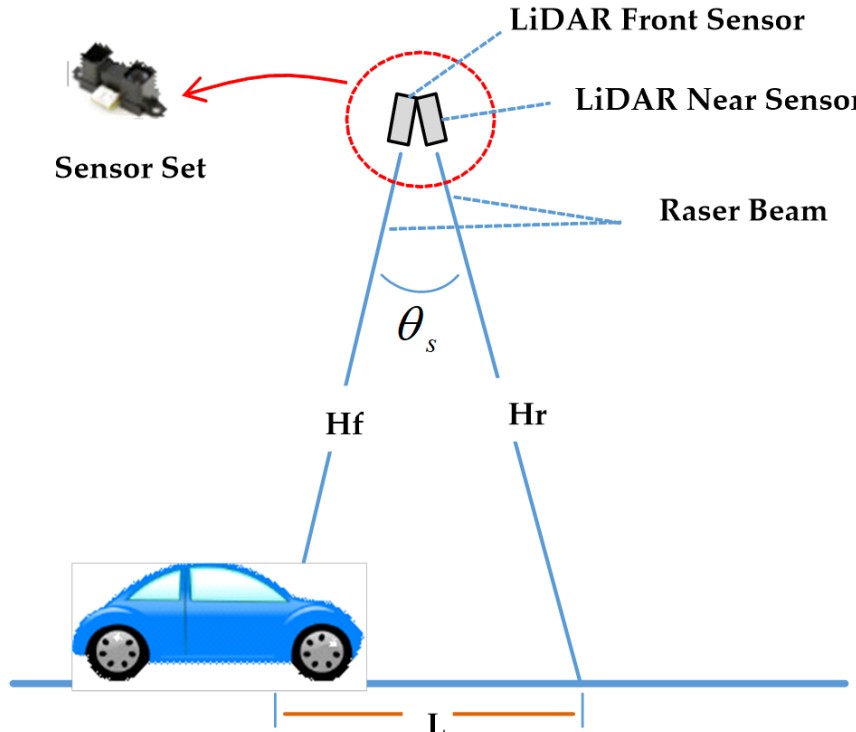

**Figure 7.** Vehicle detection on lane.

The distance between the two measuring points is expressed as follows:

$$L = \sqrt{H_f{}^2 + H_r{}^2 - \left(2\,H_f\,H_r\,cos\theta_s\right)} \tag{1}$$

From Equation (1), the driving vehicle velocity for each lane is expressed as follows:

$$V = \frac{L}{\Delta t} \tag{2}$$

In Equation (2), V is the vehicle velocity and $\Delta t$ is the measurement difference time between the LiDAR front sensor and the rear sensor at the two measuring points.

Figure 8 shows the vehicle velocity algorithm. First of all, the LiDAR front sensor measures the distance of the entry point. The distance of the $H_f$ changes when the vehicle driving through the entry measurement point. According to the change in the distance of $H_f$, it is determined that the vehicle has passed the entrance measurement point and the time of the entry measurement point is recorded. The passing time is not recorded if there is no change in the distance of the $H_f$. It records the passing time of the vehicle exit point if there is a change in $H_r$ distance when the LiDAR rear sensor is radiating the exit point. The $\Delta t$ is obtained using the passing time of the entrance measurement point of the LiDAR front sensor and the exit passing time of the LiDAR rear sensor. The passing vehicle count is determined by the change in $H_r$ measurement distance.

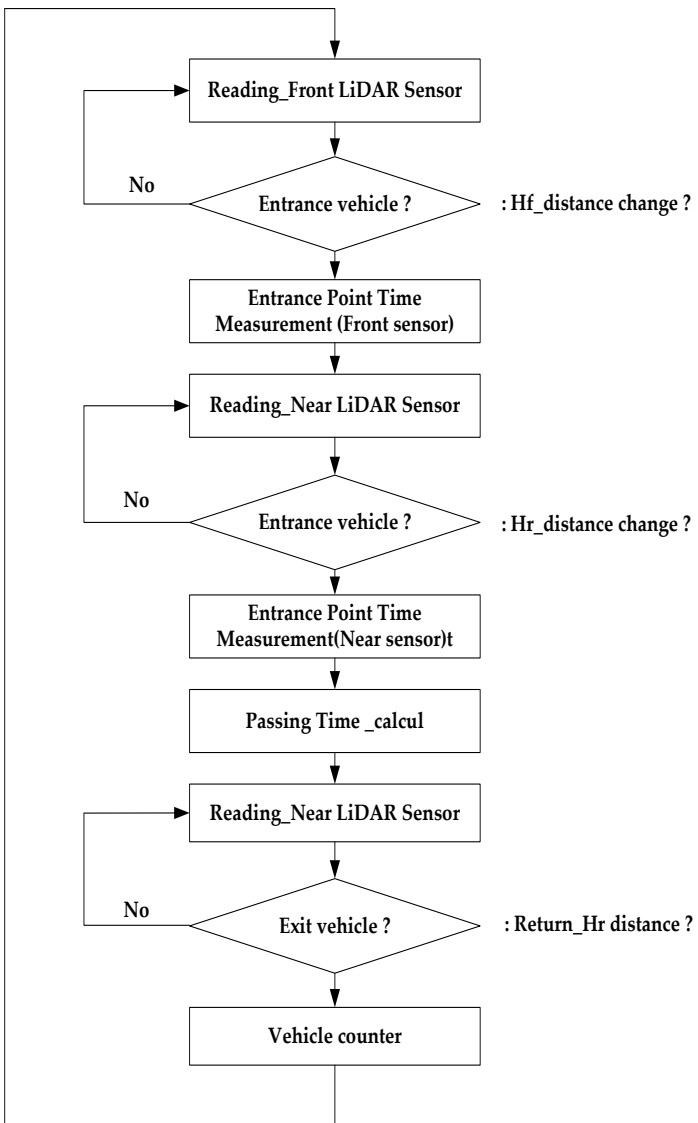

**Figure 8.** Vehicle measurement time flow chart.

## 3. Experiment of Vehicle Detection

This chapter describes an experiment to detect the driving vehicle on the road using the drone. The drone has six VDM sensor units to detect driving vehicles from the first lane to the third lane of the road, but the experiment is conducted on the first and second lanes. The reason is that the road environment for testing up to a total of three lanes is not

suitable and testing on urban roads is not safe. Thus, the test is only done up to the two entire lanes.

Figure 9a shows the octa-quad vehicle detection drone developed in this paper. The detection sensor module part is mounted on the gimbal at the bottom of the drone. Figure 9b shows the drone vehicle detection part of the detection sensor module unit and a green indicator is displayed to confirm the radiation position of the LiDAR sensor. The drone is equipped with a total of six LiDAR sensors to detect the vehicle information from lane 1 to lane 3. The radiating point's position can be checked with the green color of the indicator inside the red circle. The VDM angle can be controlled so that the LiDAR sensor can accurately radiate the measuring point of the road regardless of the drone's flying height.

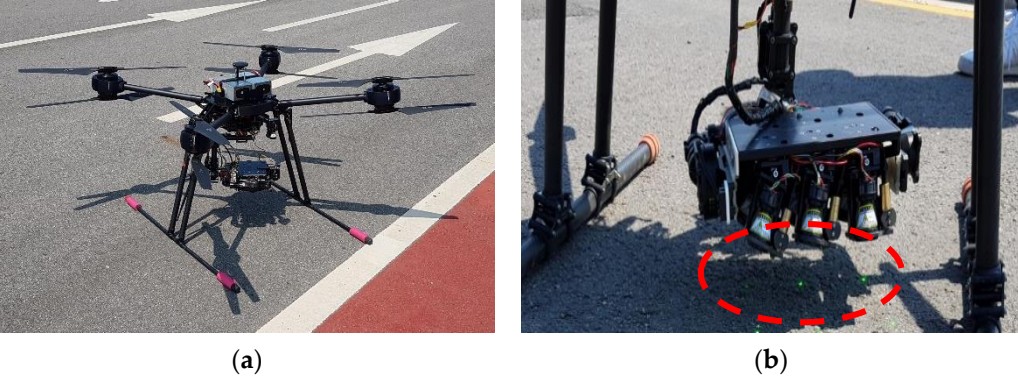

(**a**)  (**b**)

**Figure 9.** This is a developed drone for vehicle detection a lane: (**a**) drone with detection sensor part; (**b**) drone VDM indicating in the red circle.

Figure 10 shows the test environment on the road for vehicle detection. The drone's position is stopped in the vertical air outside the road and the detection module sensor unit radiates the position of the entry/exit measurement point for each lane. At this time, the indicator checks the location of the entry/exit measurement point of the LiDAR sensor. Figure 11 shows the experiment for measuring the vehicle velocity in the lane. The drone radiates the entry/exit measurement point with the LiDAR sensor to measure the vehicle's velocity in two lanes and acquires the vehicle's velocity when passing through the measurement point. VAN drives the test vehicle in the first lane and small cars in the second lane.

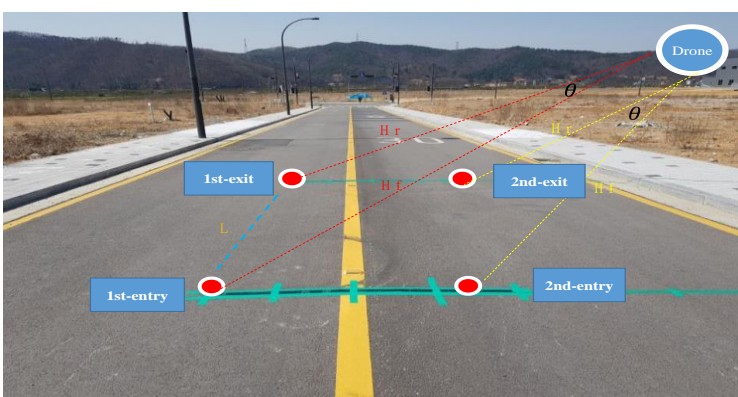

**Figure 10.** Vehicle detection experiment method on the first and second lines.

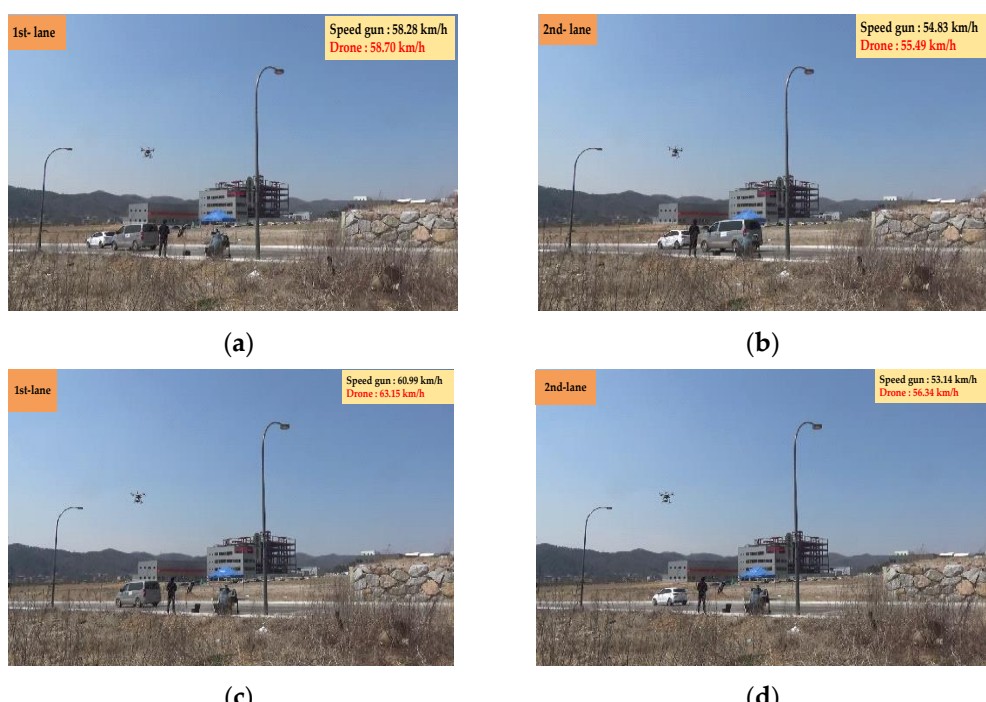

**Figure 11.** Vehicle speed experiment method on the first and second lanes; (**a**) speed gun 58.28 km/h and drone 58.70 km/h on the first lane; (**b**) speed gun 54.83 km/h and drone 55.49 km/h on the second lane; (**c**) speed gun 60.99 km/h and drone 63.15 km/h on the first lane; (**d**) speed gun 53.14 km/h and drone 56.34 km/h on the second lane.

Figure 11a,c show that the vehicle's velocity measurement in the first lane using the speed gun and the drone. Figure 11b,d show the velocity measurement of the vehicle driving in the second lane using the speed gun and the drone. Figure 11a shows 58.28 km/h and 58.70 km/h for the driving vehicle speed measurement using the speed gun and the drone, respectively. In Figure 11b, the velocity measurement of the vehicle driving the speed gun is 54.83 km/h and the measurement of the drone driving vehicle's velocity is 55.49 km/h. Figure 11c shows 60.99 km/h and 63.15 km/h for the speed gun and the drone velocity measurements, respectively. Figure 11d shows that the speed gun's driving vehicle velocity measurement is 53.14 km/h and the drone's driving vehicle velocity measurement is 56.34 km/h.

Table 1 shows the velocity of the vehicle driving in the first and the second lanes using the drone and the speed gun. The speed gun's average error and the drone in the first and the second lanes are 0.58 km/h and 1.06 km/h, respectively. The speed gun and the drone's root mean square error (RMSE) represent 0.75 km/h and 1.31 km/h in the first and second lanes, respectively. The average error probability between the speed gun and the drone is 1.2% and 2.05% in the first and second lanes, respectively.

Figure 12 shows the wind resistance test of the developed drone's ability to detect the driving vehicles. The change in wind speed is controlled by the distance between the wind turbine and the drone. The wind resistance is measured according to changes in the wind speed. Figure 12a shows the wind resistance of the drone at the wind speed of 6.6 m/s when the distance between the wind turbine and the drone is 8 m. Figure 12b shows the wind resistance of the drone in the wind speed of 7.7 m/s when the distance between the wind turbine and the drone is 8 m and Figure 12c shows wind resistance of the drone in the wind speed of 8.7 m/s when the distance is 4 m. Figure 12d shows the wind resistance of the drone in the wind speed of 8.7 m/s when the distance between the wind turbine and the drone is 2 m and Figure 12e shows the wind resistance of the drone in the wind speed of 12.1 m/s when the distance is 1 m.

**Table 1.** Comparison of the vehicle measurements for the speed gun and the drone on the multi-lane.

| No | Measurement Velocity on First Lane | | | | Measurement Velocity on Second Lane | | | |
|---|---|---|---|---|---|---|---|---|
| | Speed Gun (km/h) | Drone (km/h) | Difference | Difference Rate (%) | Speed-Gun (km/h) | Drone (km/h) | Difference | Difference Rate (%) |
| 1 | 49.49 | 50.17 | 0.68 | 1.37 | 45.48 | 42.97 | −2.51 | 5.52 |
| 2 | 52.57 | 52.32 | −0.25 | 0.48 | 48.53 | 49.63 | 1.1 | 2.27 |
| 3 | 52.32 | 52.12 | −0.20 | 0.38 | 45.44 | 46.25 | 0.81 | 1.78 |
| 4 | 43.25 | 43.63 | 0.38 | 0.88 | 46.58 | 46.37 | −0.21 | 0.45 |
| 5 | 33.00 | 34.8 | 1.8 | 5.45 | 33.28 | 33.70 | 0.42 | 1.26 |
| 6 | 47.61 | 48.09 | 0.48 | 1.01 | 49.13 | 50.11 | 0.98 | 1.99 |
| 7 | 60.99 | 63.15 | 2.16 | 3.54 | 53.14 | 56.34 | 3.2 | 6.02 |
| 8 | 56.63 | 56.99 | 0.36 | 0.64 | 47.28 | 48.53 | 1.25 | 2.64 |
| 9 | 54.83 | 55.49 | 0.66 | 1.20 | 47.15 | 47.42 | 0.27 | 0.57 |
| 10 | 55.52 | 56.45 | 0.93 | 1.68 | 58.28 | 58.70 | 0.42 | 0.72 |
| 11 | 28.02 | 28.24 | 0.22 | 0.79 | 26.43 | 27.76 | 1.33 | 5.03 |
| 12 | 44.80 | 44.48 | −0.32 | 0.71 | 45.10 | 47.05 | 1.95 | 4.32 |
| 13 | 57.75 | 57.21 | −0.54 | 0.94 | 49.21 | 49.54 | 0.33 | 0.67 |
| 14 | 54.22 | 54.11 | −0.11 | 0.20 | 45.05 | 45.27 | 0.22 | 0.49 |
| 15 | 58.32 | 59.53 | 1.21 | 2.07 | 46.90 | 45.24 | −1.66 | 3.54 |
| 16 | 68.53 | 68.68 | 0.13 | 0.24 | 68.87 | 70.97 | 2.1 | 3.98 |
| 17 | 71.64 | 72.32 | 0.68 | 1.53 | 79..34 | 80.00 | 0.66 | 1.02 |
| 18 | 75.06 | 74.81 | −0.25 | 0.52 | 87.37 | 86.47 | −0.90 | 1.38 |
| 19 | 80.13 | 79.93 | −0.20 | 0.41 | 92.65 | 93.03 | 0.38 | 0.54 |
| 20 | 84.68 | 85.06 | 0.38 | 0.86 | 100.89 | 100.00 | −0.89 | 1.24 |
| 21 | 90.35 | 91.01 | 0.66 | 0.11 | 112.38 | 113.06 | 0.68 | 0.91 |
| 22 | 98.53 | 99.00 | 0.47 | 0.68 | 125.23 | 125.98 | 0.75 | 0.99 |
| 23 | 113.58 | 113.98 | 0.40 | 0.54 | 138.85 | 139.02 | 0.17 | 0.22 |
| 24 | 127.06 | 127.39 | 0.33 | 1.70 | 150.17 | 148.57 | −1.60 | 1.50 |
| 25 | 136.85 | 138.57 | 1.72 | 2.11 | 161.35 | 159.68 | −1.67 | 2.12 |

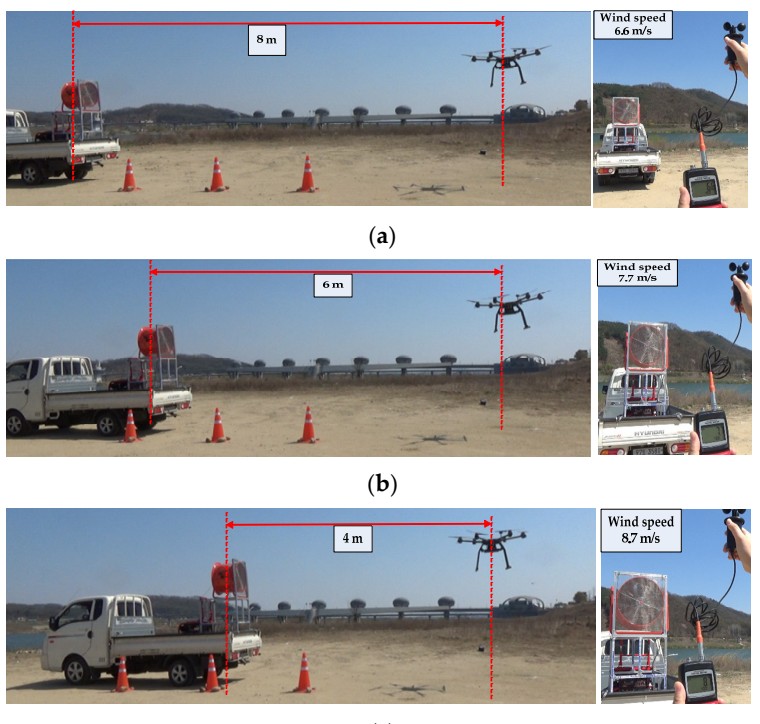

(**a**)

(**b**)

(**c**)

**Figure 12.** *Cont.*

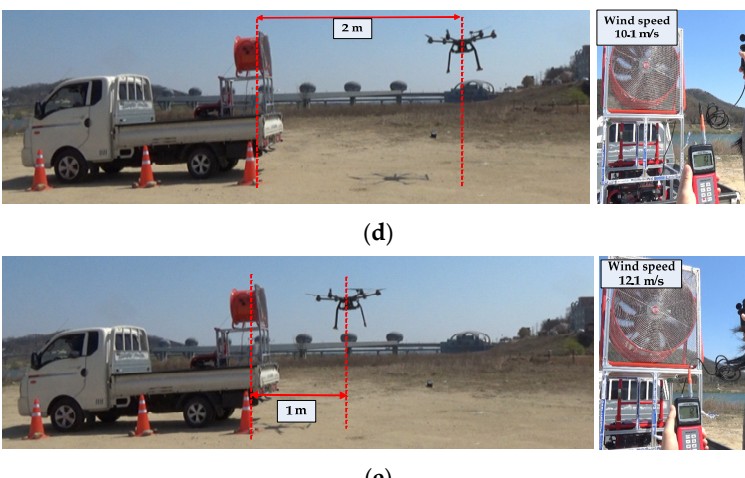

**Figure 12.** This is the wind resistance of the developed drone: (**a**) wind speed of 6.6 m/s at a distance of 8 m from the fan to the drone; (**b**) wind speed 7.7 m/s at a distance of 6 m from the fan to the drone; (**c**) wind speed 8.7 m/s at a distance 4 m from the fan to the drone; (**d**) wind speed 10.1 m/s at a distance 2 m from the fan to the drone; (**e**) wind speed 12.1 m/s at a distance 1 m from the fan to the drone.

This study detects the driving vehicle using the LiDAR sensor module mounted on the drone. It is very dangerous to conduct the experiment on the entire system on a general road without ensuring 100% safety for the drone flight. The experimental environment was not conducted on a city road, but on an outside road where no vehicle was driving. Since the drone driving vehicle detection experiment is conducted on a place without obstacles such as streetlights and electric wire, it is not possible to sufficiently consider any exceptional situations that occur on general city roads. The vehicle velocity measurement using the drone was conducted 25 times. In the experiment, the average velocity error and the total average vehicle detection rate showed performances of 1.6% and 100%, respectively. Although it was not possible to conduct the wind resistance experiment for the drone in an actual vehicle driving environment, the wind resistance of the drone was shown up to the wind speed 12 m/s, configured by an arbitrary environment.

## 4. Conclusions

In this paper, the driving vehicle was detected using the drone and the vehicle velocity was calculated based on the detection information. The driving vehicle velocity estimation was calculated from a distance and passing the time by detecting the driving vehicle with the drone LiDAR sensor at the two measurement points on the road. The drone's position for measuring the driving vehicle was stationary flying above the outside of the road, and the driving vehicle detected the first lane and the second lane at the position of the drone. The vehicle velocity estimation of the developed drone was compared with the speed gun of the existing equipment. The speed gun and drone's RMSEs were 0.75 km/h in the first lane and 1.31 km/h in the second lane. The performance of measuring the driving vehicle's velocity was similar between the drone and the existing equipment.

The existing driving vehicle velocity measuring equipment requires measuring equipment for each lane. However, the developed drone system can measure vehicles in all lanes with a single drone. Drone velocity measurement can reduce the risk of roadside traffic accidents and there is no need to install a gentry structure. In addition, vehicle information can be obtained in a dark environment. The velocity of the vehicle can be measured at a maximum wind speed of 12.1 m/s. The drone's total weight is about 12.7 kg and the hovering range is within 1 m.

This research and development (R&D) drone system has developed a driving vehicle detection system from one to three lanes, but there is no test road environment with roads from one to three lanes. Thus, the driving vehicle velocity measurement was tested in the

first and second lanes. It was impossible to test the driving vehicle's velocity in the urban road environment with up to three lanes because drone flight regulations and laws restrict it. In addition, South Korea is limited in the area and time of drone flights due to neighboring countries' environment. It is impossible to sufficiently consider the irregularities in all situations since the measurement of the driving vehicle velocity cannot be tested in various road topologies and night environments.

**Funding:** This research received no external funding.

**Institutional Review Board Statement:** Not applicable.

**Informed Consent Statement:** Not applicable.

**Data Availability Statement:** Data availability in a publicly accessible repository.

**Conflicts of Interest:** The authors declare no conflict of interest.

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
