# Peer review of "A Study on Distance Measurement Module for Driving Vehicle Velocity Estimation in Multi-Lanes Using Drones"

_applsci, doi:10.3390/app11093884_

Round 1

Reviewer 1 Report

The author dealt with a very current topic. the paper is very well organized. the abastrct is clear and precise. the paper is presented well. insert the part of the discussions before the conclusions. check the English language from the grammatical point of view. the figure 12 is missing in the text. check some typos in the text advisers to add some bibliographic sources. I suggest the following bibliographical references.

Decision Tree Method to Analyze the Performance of Lane Support Systems

G Pappalardo, S Cafiso, A Di Graziano, A Severino Sustainability 13 (2), 846    

Author Response

please refer to the attached file.

Thank you for reviewing the paper.

Reviewer 2 Report

Dear Author,

The work contains interesting preliminary results of practical research. The speed measurement method presented in the paper is understandable, however, the description of the problem / vehicle detection algorithm is unsatisfied (given the distance to the object, determining the speed is trivial). It is worth considering the description of the vehicles detection algorithm or at least provide more details about it in 2.1-2.2 (method, effectiveness, etc.)

Please answer two quite important questions related to the practical application of the developed solution:

  • How long can the presented drone work?
  • What are the operational limits of its application related to the wind? (Figure 13 shows the ability to stabilization of the drone for 12.1 m/s. Were vehicle speed tests also carried out under such conditions?)

Remarks on formatting and text editing:

  • page 2 line 5 - "Kraft" is a surname - change case of first letter,
  • do not separate the name and surname with a dot when referring to the literature (page 2, lines 26, 33, etc.),
  • formatting the list of references is necessary, please follow the MDPI style (for MDPI journals you can easily generate a citation - it is ready under each of the articles),
  • some abbreviations are not expanded (PCM, LiDAR, GPS, RGB, HDMI, BB, RMSE, GPIO, etc.) - most of them may seem obvious, but formally they should be explained.
  • there are sentences that are difficult to interpret, eg. page 12, line 3-4: "Wind speed change can change the wind speed by adjusting the wind turbine’s distance and the drone",
  • the work requires linguistic correction.

Author Response

(The authors gave the same response as above.)

Round 2

Reviewer 1 Report

changes have been made, the manuscript is a great job. it's OK for me